# The relationship between fibular head height and lower limb alignment deviation and severity after TKA for varus deformity knee osteoarthritis

Xun Qin[1,2], Hengzhi Liu[3], Yinghao Liu[1,2], Aixin Hu [1,2]*

1 The First College of Clinical Medical Science, China Three Gorges University, Yi Chang, China,
2 Yichang Central People's Hospital, Yi Chang, China, 3 Department of Orthopaedics, Huang Shi Central Hospital, Huang Shi, China

* hu_aixin@163.com

## Abstract

### Purpose

To investigate the correlation between fibular head height and the deviation and severity of lower limb alignment after TKA in patients with varus deformity and knee osteoarthritis.

### Methods

Based on the varus angle ($\gamma = 10°$), the sample was divided into two groups: < 10° and ≥ 10°, The differences in fibular head height between the two groups of patients were analyzed using an independent samples t-test; The Mann-Whitney test was used to analyze the differences in fibular head height between genders. An unordered multinomial logistic regression analysis was conducted to evaluate the effects of age, gender, height, weight, body mass index, and fibular head height on postoperative lower limb alignment; Pearson correlation analysis was used to assess the correlation between fibular head height and varus angle, preoperative HKA angle, and postoperative HKA angle; A Multiple linear regression was performed to evaluate the effects of gender, age, height, weight, body mass index, fibular head height, preoperative HKA angle, and varus angle on the postoperative HKA angle.

### Results

The sample was divided into two groups based on the varus angle: ≥ 10° and <10°. The fibular head height was $8.1825 \pm 2.72505$ mm in one group and $9.2234 \pm 2.68225$ mm in the other group ($p = 0.028 < 0.05$), which was statistically significant; The median fibular head height in females was 8.050 mm, compared to 10.645 mm in males ($p < 0.05$), which was statistically significant. In the unordered multinomial logistic regression, with postoperative lower limb alignment (varus,

**Data availability statement:** All raw data files are available from the Dryad database. Doi:10.5061/dryad.jm63xsjn7; Raw data URL: http://datadryad.org/stash/share/lvLneHDZsb-F5vZasfaX1D-2FtCuT0dvjx9E3dQsTmck.

**Funding:** The author(s) received no specific funding for this work.

**Competing interests:** The authors have declared that no competing interests exist.

valgus, neutral) as the dependent variable, In the unordered multinomial logistic regression with postoperative lower limb alignment as the dependent variable, Height, weight, and body mass index are influencing factors for postoperative lower limb alignment in patients with varus deformity knee osteoarthritis, affecting both varus and valgus alignment, while fibular head height is a significant factor for postoperative varus alignment. In the Pearson correlation analysis, fibular head height was positively correlated with postoperative HKA angle ($r = 0.212$, $p < 0.05$). In the multiple linear regression with postoperative HKA angle as the dependent variable, fibular head height and preoperative HKA angle were identified as significant factors influencing the postoperative HKA angle ($p < 0.05$).

## Conclusions

This study found that in patients with varus deformity knee osteoarthritis, a greater degree of varus was associated with a lower fibular head height. Additionally, the fibular head in female patients was positioned closer to the lateral tibial plateau compared to male patients. In varus deformity knee osteoarthritis, fibular head height is a risk factor for postoperative lower limb varus alignment following total knee arthroplasty (TKA). Patients with a higher-positioned fibular head (lower fibular head height) are more likely to develop postoperative varus malalignment after TKA. Therefore, routine measurement of fibular head height is warranted in clinical practice for patients with varus deformity knee osteoarthritis.

## Introduction

Knee osteoarthritis (KOA) is a degenerative joint disease with an increasing incidence year by year, becoming one of the major causes of disability among middle-aged and elderly individuals [1]. Statistics indicate that at least 10% of individuals aged 60 and above worldwide are affected by KOA. Clinical observations reveal that the incidence of varus deformity is more common than that of valgus deformity in primary knee osteoarthritis. Research indicates that the degenerative changes in the medial compartment of the knee joint are more pronounced than those in the lateral compartment in most KOA patients, and the risk of morbidity in the medial compartment is several times greater than that in the lateral compartment, often related to the uneven pressure distribution across the medial and lateral compartments of the knee [2].

For end-stage knee osteoarthritis, Total Knee Arthroplasty (TKA) is considered the most effective treatment [3], with the primary goal of alleviating knee pain and restoring flexion and extension function of the knee. Precise bone resection and good soft tissue balance during the procedure are key factors in achieving this goal [4,5].During the process of degenerative changes in the knee joint, changes in the osseous structure of the knee occur, such as a decrease in the proximal medial tibial angle due to collapse of the medial tibial plateau, a reduction in the angle between the distal femoral

1/3 anatomical axis and the mechanical axis of the lower limb due to changes in the osseous structure of the femoral condyles, and preoperative valgus deformity of the knee caused by knee degeneration. A study by Zhou Yi Xin et al. [6] showed that a decrease in the proximal medial tibial angle due to collapse of the medial tibial plateau leads to postoperative varus alignment of the lower limb. In a study by Wang Ben Chao on the effect of femoral deformity on postoperative lower limb alignment and knee joint line after TKA, a reduction in the angle between the distal femoral 1/3 anatomical axis and the mechanical axis of the lower limb, caused by changes in the osseous structure of the femoral condyles, significantly affected both postoperative lower limb alignment and knee joint line [7].A study by Wang Hai Hu et al. [8] found that severe preoperative valgus deformity caused by knee degeneration increases the deviation of lower limb alignment after TKA. Ma Wen Ru et al. found that lateral knee joint morphological changes occur in varus osteoarthritis, including lateral displacement of the tibial plateau, proximal fibular curvature, and upward movement of the fibular head [9]. In clinical practice, we have also observed that in patients with severe varus deformity knee osteoarthritis, the fibular head often approaches the lateral tibial plateau. Therefore, we hypothesize that changes in fibular head height in varus deformity knee osteoarthritis may affect postoperative lower limb alignment after TKA. However, this hypothesis currently lacks sufficient evidence from evidence-based medicine. Based on our review of relevant literature, we propose the following hypothesis: In patients with varus deformity knee osteoarthritis, does fibular head height affect postoperative lower limb alignment after TKA? This study included patients with varus deformity knee osteoarthritis who underwent TKA surgery at the Affiliated Hospital of China Three Gorges University (Xi ling Campus of Yichang Central People's Hospital) between June 2017 and June 2023.The objectives of this study include: ① To explore whether there are differences in fibular head height among patients with knee osteoarthritis at different levels of varus deformity; ②To investigate the correlation between fibular head height and the deviation and severity of lower limb alignment after TKA in patients with varus deformity knee osteoarthritis.

## Materials and methods

### 1. Inclusion and exclusion criteria

(1) Inclusion criteria: ① Patients hospitalized for TKA due to varus deformity osteoarthritis of the knee at our hospital;② Patients who have completed standard full-length weight-bearing X-rays preoperatively and postoperatively.

(2) Exclusion criteria: ① History of lower limb trauma;② Congenital knee deformities or skeletal dysplasia;③ Comorbidities severely affecting lower limb function, such as neurological disorders or diabetic foot; ④ History of fibular surgery;⑤ History of peripheral osteotomy around the knee (HTO or DFO);⑥ Comorbidities with other knee diseases, such as knee tumors.

### 2. Ethics statement and general information

(1) This study was reviewed and approved by the Medical and Ethics Committee of the Affiliated Hospital of China Three Gorges University,The ethical batch number was 2023-179-01. Given the retrospective nature of this study, the Ethics Committee waived the requirement for obtaining informed consent from patients, as all data were fully anonymized before access.

(2) Between June 2017 and June 2023, 185 patients with varus deformity knee osteoarthritis who visited the Orthopedics Department of the Xiling Campus of Yichang Central People's Hospital met the inclusion and exclusion criteria and underwent total knee arthroplasty (TKA).Preoperative and postoperative standard full-length weight-bearing X-rays of the lower limbs were performed, and 133 patients (133 knees) were included in the analysis. The specific patient information is as follows: 28 male cases and 105 female cases; age (67.59±7.03) years (range 53–84 years); height (159.38±6.23) cm (range 149–180 cm); weight (64.68±9.94) kg (range 42–95 kg); body mass index (25.4±3.38) kg/

m² (range 17.5–35.8 kg/m²).The measured indicators for the patients in this study are as follows: fibular head height was 9.02 mm±3.30 mm (range 2.31–18.45 mm), varus angle was 10.28°±4.57° (range 2°-29°), preoperative HKA angle was 169.50°±4.42° (range 151°-178°), and postoperative HKA angle was 176.16°±3.11° (range 161°-180°).

## 3. Date of data access

This study was a retrospective analysis conducted in July 2024, adhering to the principles of the Declaration of Helsinki.

## 4. Observation indicators

This study used the WanDong DR equipment. The patient stood with both feet on the platform in front of the long film cassette, and standing anteroposterior radiographs were taken. The X-ray beam was centered on the knee joint, with the tube at a distance of 10 feet, capturing both hips, knees, and ankles. The magnification factor of this method is typically around 5%. All observed parameters were measured five times, and the average value was taken for inclusion in the final data.

The fibular head height (H), preoperative hip-knee-ankle angle, varus angle, postoperative hip-knee-ankle angle, postoperative varus angle, and postoperative valgus angle were measured using standard full-length weight-bearing X-rays of the lower limb, with the varus angle used to assess the severity of knee varus deformity in the patients.

(1) Fibular head height is defined as the vertical distance between the highest point of the fibular head and the horizontal tangent of the lateral tibial plateau of the knee joint, reflecting the relative position of the fibular head in the vertical direction [10].A larger measured value indicates a lower position of the fibular head, whereas a smaller value indicates a higher position. On standard full-length weight-bearing X-rays of the lower limb, the vertical distance between the upper edge of the fibular head and the horizontal tangent at the lateral edge of the tibial plateau is measured (Fig 1 A).

(2) Degree of varus deformity: The hip-knee-ankle angle (HKA angle) is defined as the angle between the mechanical axis of the femur and the mechanical axis of the tibia, the angle formed by a line drawn from the center of the femoral head to the midpoint of the intercondylar notch, and a line drawn from the midpoint of the tibial plateau to the center of the ankle joint (Fig 1 B), used to assess the degree of varus deformity of the lower limb. Preoperatively, the HKA angle is measured on standard full-length weight-bearing X-rays of the lower limb.

(3) The correction angle of the preoperative HKA angle (γ) is referred to as the varus angle (Fig 1 C).In this study, we used the varus angle to assess the degree of preoperative knee varus deformity in patients, guide patient grouping, and investigate whether the height of the fibular head differs among patients with varying degrees of knee osteoarthritis and varus deformity. The patients were divided into two groups: the mild varus group (γ < 10°) and the moderate-to-severe varus group (γ ≥ 10°).The valgus deformity is shown in the figure below (Fig 1 D); the larger the valgus angle, the more severe the deformity. The measurement of the valgus angle follows the same method as the preoperative HKA angle.

(4) Postoperative Varus and Valgus Angles: Patients undergo standard weight-bearing X-ray examination of the entire lower limb within one week postoperatively. An HKA angle of less than 180° is defined as postoperative varus angle, while an HKA angle greater than 180° is defined as postoperative valgus angle. The postoperative varus angle is denoted as (α) (Fig 1 E), and the postoperative valgus angle is denoted as (β) (Fig 1 F); these two angles are complementary to the postoperative HKA angle, Both angles are correction angles of the postoperative HKA angle. These angles are used to assess whether there is lower limb alignment deviation after TKA surgery, and the lower limb alignment (varus, neutral, valgus) is analyzed statistically to determine whether the height of the fibular head is a contributing factor to lower limb alignment.

(5) Technical support: Radiographs and related data were reviewed and measured using the hospital's Picture Archiving and Communication System (PACS). A limitation is that it is difficult to determine the torsion of the tibia and fibula, as well as the rotation of the knee joint, on X-ray images.

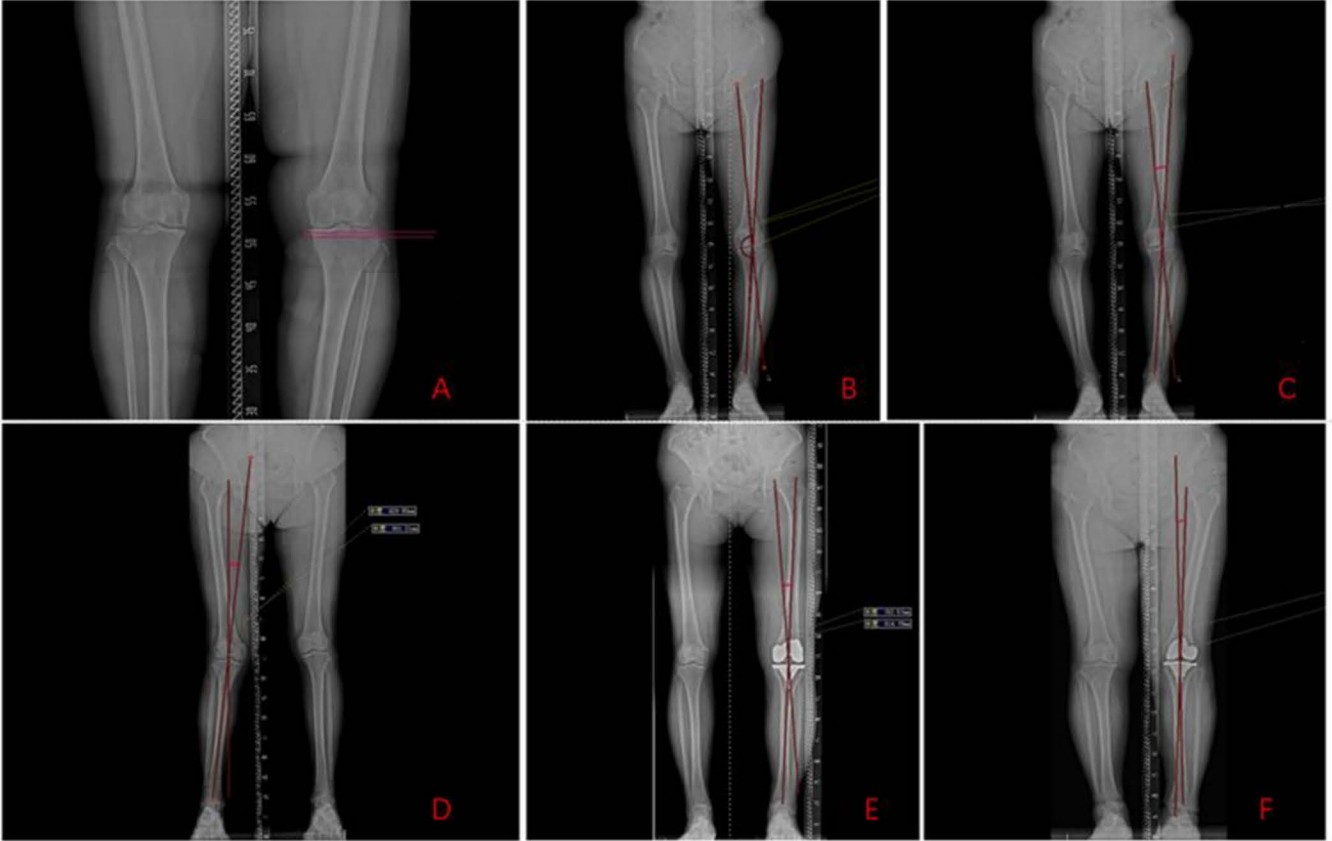

**Fig 1. Standard full-length weight-bearing X-rays of the lower limb and corresponding measurement methods.** A: Fibular head height measurement. B: Preoperative HKA angle measurement. C: Varus angle measurement. D: Valgus angle measurement. E: Postoperative varus angle measurement. F: Postoperative valgus angle measurement.

## Statistical methods

Gender, age, height, weight, body mass index, fibular head height, varus angle, preoperative HKA angle, and postoperative HKA angle are all continuous variables that conform to a normal distribution after normality testing. Gender is expressed as frequency, and intergroup comparisons are conducted using the chi-square test ($\chi^2$ test); the remaining variables are expressed as mean±standard deviation ($\overline{x}\pm s$), and intergroup comparisons are conducted using the independent two-sample t-test, with a significance level ($\alpha$) set at two-tailed 0.05.Statistical analysis was performed using the SPSS 26.0 software package.

The varus angle ($\gamma$) was divided into two groups: $\gamma \geq 10°$ with 65 cases and $\gamma < 10°$ with 68 cases. The data from the two groups were analyzed using the independent two-sample t-test、 Mann-Whitney test, with a significance level ($\alpha$) set at two-tailed 0.05.

The Mann-Whitney test was used to analyze the gender-related differences in fibular head height in the sample data, with a two-sided significance level of $\alpha = 0.05$.

Postoperative lower limb alignment (varus, valgus, neutral) was considered the dependent variable, while age, gender, height, weight, body mass index, and fibular head height were treated as independent variables, included in an unordered multinomial logistic regression analysis to explore the impact of various factors on postoperative lower limb alignment. A

95% confidence interval (CI) was used to reflect the intensity of the effect of each influencing factor on varus and valgus conditions. Additionally, Pearson correlation analysis was used to examine the correlation between fibular head height and the varus angle, preoperative HKA angle, and postoperative HKA angle, with the correlation coefficient (r) indicating the degree of correlation among these indicators. The significance level (α) was set at two-tailed 0.05.

Postoperative HKA angle was considered the dependent variable, while gender, age, height, weight, body mass index, fibular head height, preoperative HKA angle, and varus angle were treated as independent variables, conducting a stepwise multiple linear regression analysis to explore the impact of various factors on the postoperative HKA angle. The significance level (α) was set at two-tailed 0.05.

## Result

The clinical characteristics of fibular head height in the varus angle groups (γ ≥ 10° and γ < 10°) are shown in Table 1, with P < 0.05 indicating statistical significance.

The clinical characteristics of fibular head height in male and female groups are shown in Table 2. Gender shows a significant difference in fibular head height (p < 0.05). Specifically, the median value for females (8.050 mm) is significantly lower than that for males (10.645 mm). In patients with varus deformity and knee osteoarthritis, the fibular head of female patients is positioned closer to the lateral plateau of the knee joint compared to that of male patients.

Clinical characteristics of the two groups based on whether fibular head height was above or below the mean value (9.02 mm): There were 55 patients in the group with fibular head height above the mean (low fibular head group), and 78 patients in the group with fibular head height below the mean (high fibular head group). There were no statistically significant differences between the two groups in terms of age, height, weight, body mass index (BMI), or preoperative hip-knee-ankle (HKA) angle (P > 0.05). However, there were statistically significant differences in sex and postoperative HKA angle (P < 0.05, Table 3 and Table 4).

Factors Influencing Postoperative Lower Limb Alignment.

An unordered multinomial logistic regression method was applied to analyze the effects of age, gender, height, weight, body mass index, and fibular head height on postoperative lower limb alignment. The results indicated that height, weight, body mass index, and fibular head height had a significant impact on postoperative lower limb alignment (Table 5).Specifically,

**Table 1. Independent two-sample t-test with the varus angle classified as ≥ 10° and < 10°.**

|  | Sample Size | fibular head height(mm) |
| --- | --- | --- |
| Varus Angle≥10° | 65 | 8.1825 ± 2.72505 |
| Varus Angle<10° | 68 | 9.2234 ± 2.68225 |
| Statistic | – | F = 0.027 |
| P-value | – | 0.028(<0.05) |

**Table 2. Mann-Whitney test of gender as a categorical variable.**

| gender(median) | fibular head height(mm) |
| --- | --- |
| Man(n = 28) | 10.645 |
| Women(n = 105) | 8.050 |
| U-value | 1018.000 |
| Z-value | −2.495 |
| P-value | 0.013(<0.05) |

U、Z: Mann-Whitney test statistics.



**Table 3. Comparison of baseline data between the low fibular head group and the high fibular head group.**

| | Sample Size | Age ($\bar{x}\pm s$, years) | Gender (Male/Female,cases) | Height ($\bar{x}\pm s$, cm) | Weight ($\bar{x}\pm s$, kg) | Body Mass Index ($\bar{x}\pm s$, kg/m²) |
|---|---|---|---|---|---|---|
| The low fibular head group | 55 | 67.31±7.303 | 17/38 | 159.80±6.783 | 64.07±10.212 | 25.058±3.5544 |
| The high fibular head group | 78 | 67.79±6.872 | 11/67 | 158.50±8.399 | 65.10±9.756 | 25.660±3.2565 |
| Statistic | – | F=0.114 | $X^2$=5.482 | F=0.060 | F=0.010 | F=0.090 |
| P-value | – | 0.696(>0.05) | 0.019(<0.05) | 0.344(>0.05) | 0.558(>0.05) | 0.314(>0.05) |

**Table 4. Comparison of imaging measurement indices between the low fibular head group and the high fibular head group.**

| | Sample Size | Preoperative HKA Angle (°) | Postoperative HKA Angle (°) |
|---|---|---|---|
| The low fibular head group | 55 | 170.07±4.149 | 175.36±3.493 |
| The high fibular head group | 78 | 169.08±4.523 | 176.71±2.731 |
| Statistic | – | F=0.049 | F=0.634 |
| P-value | – | 0.198(>0.05) | 0.014(<0.05) |

**Table 5. Logistic regression with postoperative lower limb alignment as the dependent variable.**

| Varus and Valgus Conditions | | β | SE | P-value | OR | OR with 95% Confidence Interval | |
|---|---|---|---|---|---|---|---|
| | | | | | | Lower limit | Upper limit |
| Varus | Intercept | 162.518 | 71.997 | 0.024 | – | – | – |
| | Age | 0.033 | 0.037 | 0.362 | 1.034 | 0.962 | 1.111 |
| | Weight | 1.233 | 0.559 | 0.027 | 3.432 | 1.147 | 10.266 |
| | Height | −1.045 | 0.457 | 0.022 | 0.352 | 0.144 | 0.862 |
| | BMI | −3.046 | 1.381 | 0.027 | 0.048 | 0.003 | 0.712 |
| | 【Gender=1】 | 0.947 | 0.874 | 0.278 | 2.579 | 0.465 | 14.301 |
| | 【Gender=2】 | 0[b] | – | – | – | – | – |
| | 【h=1】 | 1.320 | 0.567 | 0.020 | 3.744 | 1.233 | 11.374 |
| | 【h=2】 | 0[b] | – | – | – | | |
| Valgus | Intercept | 234.756 | 85.466 | 0.006 | – | – | – |
| | Age | −0.001 | 0.044 | 0.985 | 0.999 | 0.917 | 1.089 |
| | Weight | 1.847 | 0.660 | 0.005 | 6.340 | 1.738 | 23.127 |
| | Height | −1.503 | 0.542 | 0.006 | 0.222 | 0.077 | 0.644 |
| | BMI | −4.520 | 1.633 | 0.006 | 0.011 | 0.000 | 0.267 |
| | 【Gender=1】 | 0.308 | 1.040 | 0.767 | 1.361 | 0.177 | 10.441 |
| | 【Gender=2】 | 0[b] | – | – | – | – | – |
| | 【h=1】 | 1.057 | 0.673 | 0.116 | 2.876 | 0.769 | 10.758 |
| | 【h=2】 | 0[b] | – | – | – | – | – |

Note: Gender: 1 indicates male, 2 indicates female; fibular head height (h): 1 indicates h>9.02, 2 indicates h<9.02;β represents the regression coefficient; SE represents the standard error of the regression coefficient; OR represents the odds ratio; CI represents the confidence interval; b represents a redundant parameter, set to 0.

the regression coefficients for weight in relation to postoperative lower limb alignment varus and valgus were 1.233 and 1.847, respectively, indicating that patients with varus and valgus alignment had a higher weight compared to those with neutral alignment. The regression coefficients for height were −1.045 and −1.503, suggesting that patients with varus and valgus alignment were shorter compared to those with neutral alignment. The regression coefficient for fibular head height in relation

**Table 6. Correlation analysis of fibular head height, varus angle, preoperative HKA angle, and postoperative HKA angle.**

|  | Fibular head height | Varus angle | Preoperative HKA | Postoperative HKA |
|---|---|---|---|---|
| Fibular head height | 1 |  |  |  |
| Varus angle | −0.078 | 1 |  |  |
| Preoperative HKA | −0.112 | 0.131 | 1 |  |
| Postoperative HKA | .212* | .171* | .393** | 1 |

Note:

*. Correlation is significant at the 0.05 level (two-tailed),

**. Correlation is significant at the 0.01 level (two-tailed).

**Table 7. Multiple Linear Regression with Postoperative HKA Angle as the Dependent Variable.**

| Indicator | β | SE | Standardized β | T-value | P-value |
|---|---|---|---|---|---|
| Constant | 126.698 | 13.762 | – | 9.206 | 0.000 |
| Fibular head height | 1.652 | 0.496 | 0.261 | 3.333 | 0.001 |
| Preoperative HKA | 0.281 | 0.056 | 0.394 | 5.037 | 0.000 |

Note: The sample size was 133; β represents the regression coefficient; SE is the standard error of the regression coefficient; the regression model $R^2$ was 0.263.

to postoperative lower limb alignment varus was 1.320, indicating that patients with varus alignment had a greater fibular head height compared to those with neutral alignment.

Factors Influencing Postoperative HKA Angle.

(1) Correlation Analysis of Imaging Measurement Indices (Table 6)

Pearson correlation analysis showed that fibular head height was positively correlated with postoperative HKA angle ($r = 0.212$, $p = 0.014 < 0.05$).

(2) Multiple Linear Regression with Postoperative HKA Angle as the Dependent Variable.

The regression model was statistically significant overall and showed good fit ($F = 6.360$, $p < 0.05$, adjusted $R^2 = 0.221$). Fibular head height and preoperative HKA angle are important influencing factors for postoperative HKA angle; Smaller fibular head height and a smaller preoperative HKA angle are significantly associated with a smaller postoperative HKA angle ($P < 0.05$).In contrast, the effects of gender, age, height, weight, body mass index, and varus angle on postoperative HKA angle were not statistically significant ($p > 0.05$, Table 7). A lower fibular head height (a higher position of the fibular head) is associated with a greater postoperative varus angle, indicating a more pronounced varus alignment after TKA and a larger varus deviation of the lower limb mechanical axis.

## Discussion

### Conclusion

- This study found that, in patients with varus deformity knee osteoarthritis, the greater the degree of varus, the lower the fibular head height; gender showed a significant difference in fibular head height.

- In varus deformity knee osteoarthritis, weight, height, body mass index, and fibular head height are factors influencing post-operative lower limb alignment after TKA, with fibular head height being a factor affecting postoperative varus alignment.

- Fibular head height is positively correlated with the postoperative HKA angle.



1. **Advantages and disadvantages of different fibular head height measurement techniques.** In this study, the measurement method for fibular head height was the vertical distance between the superior margin of the fibular head and the horizontal tangent of the lateral tibial plateau from the sagittal plane. From the sagittal plane, the fibular head is located at the posterior-lateral aspect of the tibia, and it is clearly visible on the knee's anteroposterior X-ray, which meets the measurement requirements. X-ray, as a conventional imaging diagnostic tool for knee osteoarthritis, is cost-effective and holds unique value in terms of diagnosis and patient benefit. Knee X-rays present as two-dimensional images, which cannot accurately assess the relative rotational relationship between the tibia and fibula. Furthermore, some patients with varus deformity may develop fibular osteophytes due to the impact of knee osteoarthritis. Therefore, these two factors make it difficult to completely eliminate measurement errors in two-dimensional plane measurements. In a study by Ma et al. [10], the relationship between fibular head height and the incidence and severity of varus knee osteoarthritis was explored. The measurement method involved the distance between the superior margin of the fibular head and the horizontal tangent of the lowest point of the lateral tibial plateau, which is similar to the method used in this study. The distance between the superior margin of the fibular head and the lateral tibial plateau is small in normal individuals, making this measurement method more difficult and prone to error. Additionally, some studies have used 3D CT to construct a fibular head centroid model [11], and then measured the vertical distance between the centroid and the horizontal tangent of the lateral tibial plateau. The construction of the fibular head centroid model avoids the influence of osteophytes and the issue of relative rotation between the fibula and tibia, allowing for a more accurate measurement of the distance from the fibular head centroid to the lateral tibial plateau in the three-dimensional plane. However, the cost of 3D CT imaging for full-length lower limb weight-bearing X-rays is prohibitively expensive. The aforementioned measurement methods can all be used to measure fibular head height. Combining local 3D CT imaging of the knee with standard full-length lower limb weight-bearing X-rays can comprehensively avoid measurement errors and the issue of high costs.

2. **In patients with varus knee osteoarthritis, high-positioned fibular head is a factor associated with knee joint instability.** The fibular head serves as the attachment point for several important ligaments and tendons, including the lateral collateral ligament, arcuate ligament, popliteofibular ligament, biceps femoris, and fibularis longus. Additionally, the tendon of the popliteus is indirectly connected to the fibular head via the popliteofibular ligament. Therefore, the position of the fibular head is closely related to the function of these ligaments and tendons. Biomechanical studies have shown that, compared to the normal population, patients with varus knee osteoarthritis have higher peak activity levels of the fibularis longus and lower peak activity levels of the biceps femoris. On the first day after PFO surgery, the fibular head descended, and the activity level of the fibularis longus significantly decreased. Six months post-surgery, the activity level of the biceps femoris significantly increased and remained at a higher level. Theoretically, a decrease in fibular head height would lead to the relaxation and reduction in tension of the lateral knee ligaments and tendons, which are crucial for maintaining knee stability [12].The posterolateral ligament complex of the knee, including the lateral collateral ligament, arcuate ligament, popliteofibular ligament, and popliteus tendon, generates the greatest restraining force on tibial external rotation when the knee is flexed between 30° and 45° [13].In knee flexion from 0°to 30°, the lateral collateral ligament is the primary restraint against tibial external rotation; as the knee continues to flex, the popliteus tendon complex plays a more important role [14]. Therefore, the lateral soft tissues of the knee play an important role in limiting abnormal external rotation of the tibia. This study has confirmed that, in patients with varus knee osteoarthritis, greater varus deformity is associated with a smaller fibular head height. An increase in the relative position of the fibular head may lead to a reduction in the tension of the lateral soft tissues, causing knee instability in the horizontal plane, manifesting as abnormal tibial external rotation. Previous studies have confirmed that the tibial external rotation angle in osteoarthritis patients is indeed abnormally increased [15], and that the angle increases with the severity of the arthritis [16]. In patients with varus knee osteoarthritis, a high-positioned fibular head (low fibular head height) is a factor associated with knee joint instability.

**3. In varus deformity knee osteoarthritis, fibular head height is a risk factor for postoperative lower limb varus alignment.** Lu Ji Wei et al.[17] suggested that for female patients with knee osteoarthritis, a tibial proximal bone resection thickness of 8 mm may damage the fibula and fibular collateral ligaments, while the corresponding bone resection thickness for male patients is 10 mm. Additionally, studies have shown that in patients with varus deformity knee osteoarthritis, due to the contraction of the medial ligament complex and the laxity of the lateral collateral ligament, a standard 10 mm bone resection below the lateral tibial plateau cannot be performed. The recommended tibial bone resection thickness is 7 mm, which effectively restores the posterior femoral condyle offset and joint line height, maintaining stability of the knee joint after TKA [18].The fibula is closer to the lateral tibial plateau at varying degrees of varus deformity, thus tibial proximal bone resection should be performed with greater caution. In this study, patients with a varus angle ≥10° had an average fibular head height of 8 mm, while those with a varus angle <10° had an average fibular head height of 9 mm, which is consistent with related research findings. Therefore, the procedure for tibial bone resection must be conducted more cautiously to avoid damaging the fibula and its associated ligaments. Many clinicians are relatively conservative when performing tibial bone resection [19]. During surgery, the balance between extension and flexion gaps should be assessed based on trial component placement, and the tightness at different flexion angles should be evaluated. In cases of excessive tightness, the proximal tibial bone resection should be increased, and medial soft tissue release should be performed to ensure the knee joint achieves balance between the medial and lateral sides at different flexion and extension positions. In patients with more severe varus deformity (higher preoperative fibular head position), surgeons tend to perform excessive medial soft tissue release, which may result in postoperative imbalance between the medial and lateral compartments of the knee. This imbalance could be a contributing factor to postoperative pain and may further lead to varus malalignment of the lower limb and a decreased long-term prosthesis survival rate. Therefore, the authors believe that in surgeries for varus deformity knee osteoarthritis, especially in patients with a more degree of varus deformity(a higher fibular head preoperatively), greater attention should be paid to medial soft tissue release to achieve both static and dynamic balance of the medial and lateral soft tissues. In this study, it was found that patients with varus knee osteoarthritis and a high-positioned fibular head (a smaller H value) were more likely to exhibit postoperative varus malalignment of the lower limb following total knee arthroplasty (TKA). Therefore, it is recommended that routine measurement of fibular head height be performed in patients with varus knee osteoarthritis in clinical practice. For patients with a high fibular head (a small H value), conservative tibial osteotomy should be accompanied by careful medial soft tissue release, avoiding over-release, to optimize restoration of the lower limb mechanical axis.

In recent years, kinematic alignment has been associated with postoperative varus malalignment of the lower limb following TKA. However, the authors believe that traditional mechanical alignment still retains its significance. The effectiveness of traditional mechanical alignment in TKA is widely recognized, with generally favorable outcomes and a prosthesis survival rate exceeding 90% at 15 years [20]. Kinematic alignment advocates for preserving the native anatomy of the knee by avoiding ligament release and complete correction of preoperative varus, valgus, or flexion deformities [21]. Although kinematic alignment allows for a degree of postoperative varus alignment, in patients with severe preoperative varus osteoarthritis, excessive tightness of the medial soft tissues can significantly compromise postoperative limb alignment. Technological advancements, including patient-specific cutting guides, computer-assisted navigation systems, and orthopedic robotics, have been introduced to support kinematic alignment in TKA, showing clear advantages over traditional mechanical techniques. However, in resource-limited settings, traditional mechanical alignment remains the predominant approach. Therefore, the authors believe that the value of mechanical alignment should not be underestimated.

In summary, The lateral morphological changes in the knee joint occur due to varus deformity of the knee, including upward displacement of the fibular head. In other words, in patients with severe varus deformity of the knee, the fibular head is very close to the tibial plateau. The height of the fibular head is positively correlated with the postoperative HKA angle; low fibular head height High-positioned fibular head) can lead to knee instability. Therefore, patients with varus deformity of the knee and a higher fibular head (low fibular head height) are more likely to develop lower limb varus

deformity after total knee arthroplasty. For patients with varus deformity of the knee and a higher fibular head, a smaller range of tibial osteotomy should be performed, paying particular attention to medial soft tissue release, do not perform extensive release, aiming to restore the alignment of the lower limb as much as possible. Finally, in clinical practice, it is necessary to routinely measure the height of the fibular head in patients with varus deformity of the knee and osteoarthritis.

## Supporting information

**S1 Data. The data are the original measurement data used in this study and do not contain patient privacy information.**
(ZIP)

## Acknowledgments

Thank Dr. Hu for guiding the research direction, Hengzhi Liu for participating in the original data analysis and verification, and Yinghao Liu for participating in the original data collection.

## Author contributions

**Conceptualization:** AiXin Hu.

**Data curation:** YingHao Liu.

**Formal analysis:** HengZhi Liu.

**Methodology:** HengZhi Liu.

**Supervision:** AiXin Hu.

**Writing – original draft:** Xun Qin.

**Writing – review & editing:** Xun Qin.

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
