## [Decision Letter · Decision Letter 0]

23 Dec 2024

PONE-D-24-52473Correlation between fibular head height and residual varus and valgus deformities after TKA for varus osteoarthritisPLOS ONE

Dear Dr. Hu,

Thank you for submitting your manuscript to PLOS ONE. After careful consideration, we feel that it has merit but does not fully meet PLOS ONE’s publication criteria as it currently stands. Therefore, we invite you to submit a revised version of the manuscript that addresses the points raised during the review process.

We look forward to receiving your revised manuscript.

Kind regards,

Sina Afzal

Academic Editor

PLOS ONE

2. In this instance it seems there may be acceptable restrictions in place that prevent the public sharing of your minimal data. However, in line with our goal of ensuring long-term data availability to all interested researchers, PLOS’ Data Policy states that authors cannot be the sole named individuals responsible for ensuring data access (http://journals.plos.org/plosone/s/data-availability#loc-acceptable-data-sharing-methods).

5. We note that there is identifying data in the Supporting Information file <Raw data and data analysis.zip>. Due to the inclusion of these potentially identifying data, we have removed this file from your file inventory. Prior to sharing human research participant data, authors should consult with an ethics committee to ensure data are shared in accordance with participant consent and all applicable local laws.

-Location data

Please remove or anonymize all personal information, ensure that the data shared are in accordance with participant consent, and re-upload a fully anonymized data set. Please note that spreadsheet columns with personal information must be removed and not hidden as all hidden columns will appear in the published file.

Reviewers' comments:

Reviewer's Responses to Questions

**Comments to the Author**

1. Is the manuscript technically sound, and do the data support the conclusions?

Reviewer #1: No

Reviewer #2: Partly

2. Has the statistical analysis been performed appropriately and rigorously? 

Reviewer #1: No

Reviewer #2: I Don't Know

3. Have the authors made all data underlying the findings in their manuscript fully available?

Reviewer #1: No

Reviewer #2: No

4. Is the manuscript presented in an intelligible fashion and written in standard English?

Reviewer #1: No

Reviewer #2: No

5. Review Comments to the Author

Reviewer #1: Title

It did not emphasize the highlights.

the term "Residual" is generic. It can be improved.

Abstract

It can be shortened.

The purpose is unclear. "Residual" need to be defined.

In the methods, residual valgus was not specified clearly. It can be improved.

The tibial rotation related with fibular height was not mentioned. Why not?

Conclusion

It can be shortened. It is different from the main text.

The clinical relevance remains uncertain. It can be improved

Key words.

Some terms did not match in the MeshTerms. It is suggested to check for proper terminology.

Introduction

It can be shortened.

The justification and contextualization were not stated clearly.

There are some unespecific terms like: Row 86: more common; Row 88: more pronounced; Row 97: varying degrees, Row 99: greater angle; Row 100: severe varus; Row 116: greater varus severity; row 117: significant differences

It is lacking information about anatomic variation for fibular height and morphology, mainly between male and female It can be improved.

The classification for varus and valgus deformities were not mentioned. Why not?

The different techniques for fibular height were not written. It can be impoved.

The hypothesis was not written.Why not?

The purpose was not stated clearly. Row 132: What it means "significant difference"? statistically or clinically? It can be improved.

Row 134"the residual varus or valgus were not determined with numbers.It can be improved.

Methods:

The end points were not described clearly

The tibial torsion and knee rotational were not described. It is suggested to describe with more details

The sagital view for varus and valgus deformity were not evaluated. Why not?

It is suggested to add references for varus and valgus deformity, including residual varus and valgus.

The anatomic variation forfibular height and tibial anatomy, mainly between male female were not written. It is sugested to describe in more details.

The techinque used in the X-Ray wasnot written in details. How was determined the magnification and rotational problems?

Results

The statitical X clinical relevance were not described clearly. It can be improved.

THe morphotype of the participants were not written. Why not?

The accuracy of the measurements were not determined. Why not?

Discussion

THe main findings were not correlated with clinical scenario. it can be improved.

The comparison among the technique used in this study could be more explored in this section.

The clinical relevance remains unclear.

Conclusion.

It can be shortened according with main findings and goal.

It is different from the abstract

Refrences OK

Reviewer #2: This manuscript contains numerous English errors that require correction before submission. For example, "steoarthritis" in line 129 should be corrected to "osteoarthritis," and "orrelation" in line 133 should be corrected to "correlation."

In line 95, I am uncertain if "Osteotomy" is the most appropriate term for Total Knee Arthroplasty (TKA). Most publications utilize "bone resection" in this context.

The presentation of results in the introduction section (lines 117-119: "We also found significant differences in the distance between the fibular head and the lateral plateau of the knee among patients with varying degrees of varus deformity") is unusual and should be relocated to the appropriate results section.

I am curious about the approximately equal distribution of patients between the cohorts (more and less than 10 degrees). Was this allocation randomized or accidental?

6. PLOS authors have the option to publish the peer review history of their article (what does this mean? ). If published, this will include your full peer review and any attached files.

**Do you want your identity to be public for this peer review?** For information about this choice, including consent withdrawal, please see our Privacy Policy .

Reviewer #1: No

Reviewer #2: No

---

## [Author Response · Author response to Decision Letter 1]

21 Jan 2025

Letters of reply to reviewers and editors are attached to Attach Files.

---

## [Decision Letter · Decision Letter 1]

24 Mar 2025

PONE-D-24-52473R1The relationship between fibular head height and lower limb alignment deviation and severity after TKA for varus deformity knee osteoarthritisPLOS ONE

Dear Dr. Hu,

Thank you for submitting your manuscript to PLOS ONE. After careful consideration, we feel that it has merit but does not fully meet PLOS ONE’s publication criteria as it currently stands. Therefore, we invite you to submit a revised version of the manuscript that addresses the points raised during the review process.

We look forward to receiving your revised manuscript.

Kind regards,

Sina Afzal

Academic Editor

PLOS ONE

Reviewers' comments:

Reviewer's Responses to Questions

**Comments to the Author**

1. If the authors have adequately addressed your comments raised in a previous round of review and you feel that this manuscript is now acceptable for publication, you may indicate that here to bypass the “Comments to the Author” section, enter your conflict of interest statement in the “Confidential to Editor” section, and submit your "Accept" recommendation.

Reviewer #3: All comments have been addressed

Reviewer #4: (No Response)

2. Is the manuscript technically sound, and do the data support the conclusions?

Reviewer #3: Yes

Reviewer #4: Partly

3. Has the statistical analysis been performed appropriately and rigorously? 

Reviewer #3: Yes

Reviewer #4: N/A

4. Have the authors made all data underlying the findings in their manuscript fully available?

Reviewer #3: Yes

Reviewer #4: Yes

5. Is the manuscript presented in an intelligible fashion and written in standard English?

Reviewer #3: Yes

Reviewer #4: Yes

6. Review Comments to the Author

Reviewer #3: Lateral morphological changes of the knee joint occur with varus deformity of the knee joint and include upward migration of the fibular head. In other words, in patients with severe osteoarthritis of the knee, the fibular head is very close to the tibial plateau. This height of the fibular head, which contributes to tibial torsion and causes knee instability, was positively correlated with the postoperative HKA angle. Thus, patients with a low fibular head are more prone to lower extremity internal rotation after TKA. Therefore, patients with a low fibular head should undergo a less extensive tibial osteotomy, paying particular attention to medial soft tissue release and restoring as much lower extremity alignment as possible. In conclusion, it is recommended that patients with a low fibular head undergo proximal tibial osteotomy using the intraosseous alignment technique, which more effectively restores postoperative lower extremity alignment. Is this an acceptable conclusion? If so, please add a further concluding paragraph as short and concise as this one.

Furthermore, the recent concept of kinematic alignment allows for varus deformity after TKA. If this is the case, the above caution would no longer be necessary, but I thought the paper would be better if the author's thoughts on this point were also added somewhere.

Reviewer #4: First of all I appreciate the effort made by the authors in this original manuscript

If this paper is to be accepted there are some issues to be elaborated by the authors

First, I have difficulty in agreeing that the height of the fibular head is a contributing factor in arthritis and knee coronal plane deformity

The height can affect the stability surely by the contribution of the fibular head the attachment of the lateral stabilizing structures but the femorotibial joint alignment is affected by the collapse of the femoral or tibial condyles as mentioned by the authors in the introduction.

The fibula here is extraarticular structure and the fibular height before or after surgery and its relation to the coronal plane alignment is merely observational and secondary to the original alignment as long as there is no lateral major instability

Second. the choice of the cutoff value determined as 10 degrees of varus aligment and its use in statistical comparison.

Third, the conclusions and recommendations from this study should be revised as for example

The finding that patients with a lower fibular head were more likely to experience postoperative varus alignment of the lower limb after total knee arthroplasty (TKA) and therefore, it is essential to routinely measure fibular head height in patients with varus deformity knee osteoarthritis in clinical practice.

Also the recommendation of the choice of intramedullary or extramedullary alignment method , the authors declare that it is recommended to use the intramedullary alignment method during surgery for proximal tibial bone resection. This is controversial as well.

7. PLOS authors have the option to publish the peer review history of their article (what does this mean? ). If published, this will include your full peer review and any attached files.

**Do you want your identity to be public for this peer review?** For information about this choice, including consent withdrawal, please see our Privacy Policy .

Reviewer #3: **Yes: ** Hiroaki Kijima

Reviewer #4: No

---

## [Author Response · Author response to Decision Letter 2]

11 Apr 2025

The opinions of all reviewers have been analyzed and explained.

---

## [Decision Letter · Decision Letter 2]

10 June 2025

The relationship between fibular head height and lower limb alignment deviation and severity after TKA for varus deformity knee osteoarthritis

PONE-D-24-52473R2

Dear Dr. Hu,

We’re pleased to inform you that your manuscript has been judged scientifically suitable for publication and will be formally accepted for publication once it meets all outstanding technical requirements.

Kind regards,

Ahmed A. Khalifa, M.D., FRCS, MSc.

Academic Editor

PLOS ONE

Additional Editor Comments (optional):

Thanks for the effort.

Reviewers' comments:

Reviewer's Responses to Questions

**Comments to the Author**

1. If the authors have adequately addressed your comments raised in a previous round of review and you feel that this manuscript is now acceptable for publication, you may indicate that here to bypass the “Comments to the Author” section, enter your conflict of interest statement in the “Confidential to Editor” section, and submit your "Accept" recommendation.

Reviewer #3: (No Response)

Reviewer #4: All comments have been addressed

2. Is the manuscript technically sound, and do the data support the conclusions?

Reviewer #3: (No Response)

Reviewer #4: Yes

3. Has the statistical analysis been performed appropriately and rigorously? 

Reviewer #3: (No Response)

Reviewer #4: Yes

4. Have the authors made all data underlying the findings in their manuscript fully available?

Reviewer #3: (No Response)

Reviewer #4: (No Response)

5. Is the manuscript presented in an intelligible fashion and written in standard English?

Reviewer #3: (No Response)

Reviewer #4: Yes

6. Review Comments to the Author

Reviewer #3: (No Response)

Reviewer #4: The revision is satisfactory

The discussion and conclusions are well-written

Please rearrange to put the conclusion in the final part after discussion

7. PLOS authors have the option to publish the peer review history of their article (what does this mean? ). If published, this will include your full peer review and any attached files.

**Do you want your identity to be public for this peer review?** For information about this choice, including consent withdrawal, please see our Privacy Policy .

Reviewer #3: No

Reviewer #4: No

---

## [Editor Report · Acceptance letter]

PONE-D-24-52473R2

PLOS ONE

Dear Dr. Hu,

I'm pleased to inform you that your manuscript has been deemed suitable for publication in PLOS ONE. Congratulations! Your manuscript is now being handed over to our production team.

Kind regards,

on behalf of

Dr. Ahmed A. Khalifa

Academic Editor

PLOS ONE